# PLANES: Plausibility analysis of epidemiological signals

**V.P. Nagraj**◉*, **Amy E. Benefield, Desiree Williams, Stephen D. Turner**◉

Signature Science, LLC, Charlottesville, Virginia, United States of America

* pnagraj@signaturescience.com

## Abstract

Methods for reviewing epidemiological signals are necessary to building and maintaining data-driven public health capabilities. We have developed a novel approach for assessing the plausibility of infectious disease forecasts and surveillance data. The PLANES (**PL**ausibility **AN**alysis of **E**pidemiological **S**ignals) methodology is designed to be multi-dimensional and flexible, yielding an overall score based on individual component assessments that can be applied at various temporal and spatial granularities. Here we describe PLANES, provide a demonstration analysis, and discuss how to use the open-source rplanes R package. PLANES aims to enable modelers and public health end-users to evaluate forecast plausibility and surveillance data integrity, ultimately improving early warning systems and informing evidence-based decision-making.

## Introduction

Near-term forecasts and long-term projections of infectious disease targets are critical to effective public health communication, decision-making, and resource allocation. Review of surveillance data and model output prior to dissemination is paramount for engendering trust in model-based policy and public health decision-making. Real-time efforts to model Ebola [1], COVID-19 [2], and influenza [3] have shown that even well-calibrated methods may yield implausible trajectories for certain targets. For consortia that openly solicit forecast contributions, implausible submissions could bias ensembles so as to misguide policymaking or erode confidence in public health communication. Likewise, some surveillance systems may suffer from delayed or temporarily faulty reporting mechanisms. Assessing plausibility of surveillance data in real-time could mitigate the impact of data integrity issues on forecasting efforts and identify systematic problems to fix, both of which would contribute to more effective early warning systems. However, there are currently no codified plausibility heuristics, and if any plausibility assessment of forecasts or surveillance signals is performed at all, it is typically *ad hoc* and results are not broadly disseminated [4,5].

To address the gap in methodology for reviewing epidemiological signals, we developed a novel approach, PLANES (Plausibility Analysis of Epidemiological Signals), for flexible, multi-dimensional assessment of near-term forecasts and surveillance data integrity. We conceived of the PLANES algorithm as a method to provide an overall score based on individual component assessments. The PLANES scoring mechanism was designed to be agnostic to temporal and spatial granularity. We delivered the PLANES methodology via an open-source

**Data availability statement:** The data underlying the results presented in the study are available from FluSight (https://github.com/cdcepi/Flusight-forecast-data).

**Funding:** This work was supported by a subaward to Signature Science, LLC from CSTE via CDC Cooperative Agreement No. NU38OT000297. There was no additional external funding received for this study."

**Competing interests:** The authors have declared that no competing interests exist.

software package such that the approach could be readily operationalized by modelers and public health end-users working across federal, state, territorial, local, and tribal jurisdictions in the United States and elsewhere.

## Methods

### PLANES algorithm

Assessing whether a reported or forecasted value is plausible requires a baseline against which the signal can be compared. We designed the PLANES algorithm to use historical observed data deemed trustworthy to create initial "seed" characteristics to assess plausibility downstream. Each seed characteristic is location-specific such that plausibility analyses independently assess multiple locations in the forecast or surveillance data. One of the priorities in designing the PLANES methodology was the inclusion of multiple components. There are existing approaches for one-dimensional anomaly detection [6], and we therefore sought to develop an approach to examine multiple features of the data. The algorithm uses the relevant seed characteristics for assessments of individual components, each of which deliver a binary determination of whether the data appear implausible. Collectively, the binary assessments can be summarized to deliver an ordinal score.

It is important to note that the scoring system we envisioned would work for both forecasts and observed signals. However, we anticipated that certain components may include characteristics that would only be applicable to forecasted signals. We also expected that the presence of flags raised for certain components may be more or less of interest for specific use-cases, and therefore aimed to accommodate a weighting scheme to allow users to modulate the impact of components in the overall score. S1 Fig demonstrates the concept behind the algorithm, with illustrations of notional components and impacts.

### R package

To deliver the PLANES algorithm, we developed an open-source R package [7]. The package, rplanes, was scoped to include functions to prepare and format the data for analysis, create the background seed characteristics, and run the plausibility analysis. We aimed to make the package as user-friendly as possible, with helpers to intuitively prepare data and a wrapper function to run all individual components. As part of the package, we also designed an interface to translate the programmatic API to point-and-click features via Shiny [8].

### Demonstration analysis

After implementing PLANES in the rplanes package, we demonstrated how the approach can be used operationally by applying the plausibility scoring on real-world data. For this effort we used the 2022-23 FluSight forecasts [9]. We retrospectively masked the available weekly observed data to generate the background seed characteristics. We then iteratively ran the plausibility scoring to create a distribution of PLANES scores across the time span and signals analyzed. All weekly plausibility analyses were stratified by location and submitting forecaster.

Beyond a demonstration of the package features, our analysis aimed to 1) verify that the rplanes functionally worked when using operational data, 2) assess the sensitivity of the components in the algorithm (i.e., distribution of how many flags were raised), 3) evaluate the correlation between PLANES and forecast performance, and 4) study the impact of different weighting schemes and varying component sets on the PLANES score.

For the functional testing, we set out to confirm that the rplanes functions could be used without any errors during processing. When assessing the sensitivity of the PLANES algorithm, we aggregated all scores generated and counted how many times each score was

observed. To correlate the PLANES scores with forecast performance, we used the weighted interval score (WIS) to estimate the accuracy of predictions [10]. As with the PLANES scores, the WIS was calculated across strata of forecast week, location, and forecaster. The WIS was further stratified by forecast horizon, but for this analysis we computed a mean WIS across all horizons to align with the resolution of the PLANES score. To demonstrate the impact of weighting schemes, we calculated the PLANES scores across all forecasts in the season using 100 randomly sampled weighting schemes. Each weighting scheme was defined as a vector with seven elements, each of which was drawn from one through seven without replacement. We also calculated the PLANES scores with equal weights using each of the 127 possible combinations of the seven available components.

## Data

**FluSight forecasts.**  To assess the plausibility of a forecast signal, we analyzed forecast submissions to the FluSight hub from the 2022-23 season [11]. During this season, the FluSight coordinators solicited weekly forecasts of incident flu hospitalizations in the United States. Participating teams were allowed to submit forecasts using multiple models. Here we refer to each combination of team and model as a forecaster. Submissions included required metadata for the team name, contact information, model designation (i.e., "primary", "secondary", "proposed", or "other"), and licenses for forecasts. For this analysis we excluded forecasters that were designated as proposed or other, as well as any forecasters that had an ambiguous or non-permissive license. All forecasts were retrieved from the public GitHub repository for the challenge.

## Results

### Components

We developed a set of components to assess plausibility of epidemiological signals. Individually, each component provides a binary assessment (i.e., yes/no is the signal implausible). All evaluated components are then combined into an ordinal score. By default, each component is equally weighted in the overall PLANES score. When delivered in the rplanes R package, the user can optionally weight components higher or lower in the scoring scheme. What follows is a description of the characteristics assessed and methods used for each of the seven PLANES components we implemented. For each component we describe the internal logic and basic motivation. Examples of each component are visually depicted and detailed in equations where applicable in S1 File.

### Difference

The difference component checks the magnitude of point-to-point differences for all time steps of the evaluated data. This component can be used on either forecasts or observed signals. If an evaluated signal departs from the prior observation more dramatically than has been seen previously in the time series, then it is flagged as implausible. The function internally computes the maximum observed difference (based on absolute value) to set a threshold, which if exceeded will trigger a flag to be raised by the algorithm. While large and unexpected point-to-point changes may naturally occur in epidemiological signals, this component provides a means to draw attention to the most extreme cases.

### Cover

The coverage component compares the prediction interval for the first horizon of the evaluated signal to the most recent value in the seed. If the interval does not cover the most recent

data point, then the flag is raised as implausible. The width of the interval used for this evaluation can be customized by the user when preparing the signal data (see the rplanes section below for more details on the API). The narrower the width of the prediction interval, the more sensitive this component will be. This component is motivated by an expectation that the prediction interval for a poorly calibrated forecast may exhibit immediate departure from the most recent historical signal (i.e., in the first horizon), and that such implausibility would manifest in the prediction intervals as well. Note that because this component requires a prediction interval, it can only be used to assess plausibility of forecast signals.

### Taper

The taper component checks whether the prediction interval for the evaluated signal decreases in width (i.e., certainty increases) as horizons progress. Because this component requires a prediction interval, it can only be used to assess plausibility of forecast signals. The width of the prediction interval at every horizon is assessed against the previous horizon and if any of the intervals for the earlier horizon is wider a flag is raised. One would expect that there would be more variability in signals forecasted further out in time, and therefore the prediction interval would be wider in later horizons. The goal of this component is to assess this phenomenon and to flag situations where signals exhibit behavior that counters this expectation.

### Repeat

The repeat component checks whether consecutive values in an observed or forecasted signal are repeated more than the tolerated number of times ($k$). When the seed is created, it stores the maximum number of consecutive repeats for each location and uses this as the default value for $k$. If the evaluated data exceeds $k$, then the signal is considered implausible, and a flag is raised. Here a repeat is defined as two or more consecutive values that are exactly equivalent. This definition means that the component may be most informative for signals that communicate counts (e.g., number of positive influenza cases) instead of other measures (e.g., percentage of influenza-like illness visits). By default, the length of the values used to check repeats is the number of evaluated horizons plus a prepend length, which is set to the defined maximum number of repeats. The tolerance and prepend parameters can be overridden by the user in the rplanes package.

### Trend

The trend component assesses if there is a significant change in the magnitude or direction of the slope for the evaluated signal compared to the most recent data in the seed. Each "change point" in the signal is identified using a hierarchical divisive algorithm originally implemented in the ecp R package [12]. The input for the algorithm is the lagged difference of all points in a combined time series that concatenates every value in the observed and evaluated signals. All change points in the time series are identified, but a flag is only raised if there is a change point in any of the evaluated horizons or the most recent observed value. The analysis requires at least four times as many observed values as there are evaluated values. The trend component is only available for forecast signals.

As noted above, the change point analysis uses the ecp package, and specifically the e.divisive() function. The methods for the change point detection have been previously published [13]. In brief, change points are identified based on distances between segments, with larger distances indicating higher likelihood of a change point. Internally, the function uses a permutation test to calculate an approximate p-value that is used in hypothesis testing. Only statistically significant change points are flagged, and the significance level ($\alpha$) can be customized

to the use-case. A higher value for α will decrease the sensitivity of the change point detection and therefore reduce the number of trend flags raised.

## Shape

While the trend component scans the time series for an inflection point, the shape component assesses the time series for unusual shapes across multiple points. To arrive at the shape assessment, the algorithm first divides the observed seed data into sliding windows to form trajectories. The trajectories are summarized as a set of shapes against which the forecasted trajectory is compared. If the shape of the forecasted trajectory does not match any shapes in the seed data, then the forecast is considered implausible per this component. The core intuition underlying this component is that the shape of future data is more likely to reflect patterns that have previously been observed and less likely to be a novel trajectory. Therefore, it may be useful to flag any novel shapes for review.

We developed two methods for summarizing the shapes of signal trajectories. The first method uses a dynamic time warping (DTW) technique to return the Euclidean distance between sets of consecutive values. Each set is constructed by sliding across the time series of observed values for the given location in the seed data. The size of each window (i.e., length of sliding time series) is fixed to equal the number of horizons in the forecast to be evaluated. DTW methods and applications have been described in detail previously [14]. We used DTW as implemented in the dtw R package and the dtwDist() function with default parameters [15]. Our algorithm finds the minimum distances for each window, and the maximum of the minimum distances serves as a threshold. The algorithm then calculates the DTW distances between the forecast signal and every observed sliding window. Note that as part of this procedure, the algorithm builds trajectories for the forecast point estimates as well as upper and lower bounds of the prediction interval. If the distance between the forecast trajectories and any observed sliding window is less than or equal to the threshold defined above, then this shape is not considered novel.

The distance calculations in the DTW approach can be time consuming, especially as the number of observations in the seed data increases. To mitigate the computational expense, we developed an alternative approach for identifying shapes. This method uses differences of consecutive observations to construct trajectories. Each point-to-point difference is computed and then centered and scaled by standard deviation around the mean of all differences. As with DTW, we define the trajectories in sliding windows. For scaled differences, we further categorize each difference as an "increase", "decrease", or "stable" change. The threshold for increase or decrease is a difference of one standard deviation in the respective direction. Collectively, the categorical summaries of the differences within the given window form a shape (e.g., "increase;stable;stable;-decrease"). We then assess categorical changes for forecasts and compare to the set of observed shapes. If the shape is novel, then a flag is raised for implausibility. Given its computational efficiency compared to DTW, the scaled difference method is set as the default in rplanes.

## Zero

The zero component was designed to check if there are any "sudden" zeros in the evaluated signal. Whether it is a broken surveillance instrument or miscalibrated forecast, we expect it would be unlikely to observe a zero if it has never been reported in the seed data. This algorithm first identifies whether any values in the evaluated signal are equal to zero. If zeros are found, it examines the seed for the presence of any zeros. If zeroes exist in the seed, the function determines that the evaluated zero is plausible. However, if no zeroes are present in the seed, the function deems the evaluated zero implausible.

## rplanes

The PLANES algorithm is implemented in the rplanes R package. The package is published under an open-source license and is available on GitHub and the Comprehensive R Archive Network (CRAN) [16]. The package changelog, function documentation, and narrative user guides are delivered in a publicly available website (https://signaturescience.github.io/rplanes/). The package API is designed to provide an intuitive and efficient set of functions to prepare and run the PLANES analysis. The entire PLANES analysis workflow is depicted in Fig 1.

At a high level, there are three steps to the PLANES analysis with rplanes. First, the user determines if the analysis will be assessing an observed or forecasted signal. Both options are available in the to_signal() constructor function, which creates an S3 object with a primary class "signal" and a secondary class corresponding to the type of signal specified (i.e., "observed" or "forecast"). Inputs for to_signal() must be provided as a data frame, with data at daily, weekly, or monthly resolution. The input data frame requires that specific features are present depending on whether it contains observed or forecast data. For observed signals, it should include columns for location (geographic unit such as FIPS code) and date (date of reported value), along with an outcome column. For forecast signals, the data frame should include columns for location, date (corresponding to the forecast horizon), horizon, lower and upper limits of the prediction interval, and point estimates. Note that rplanes includes a helper function called read_forecast() to convert the quantile format that has been standardized in disease forecasting hubs directly from a file. Downstream analysis functions have select data validation checks in place and will issue warnings or errors for incomplete or incompatible data. However, ideally data prepared as a signal object should be cleaned (e.g., location names disambiguated) and complete (e.g., free of large gaps) prior to using to_signal().

Once the signal to be evaluated has been prepared, the user then needs to retrieve and prepare the observed data that will be used to generate baseline seed characteristics. This observed data should also be prepared with to_signal() before being passed to plane_seed(), which returns a named list with summarized characteristics for each of the locations in the dataset. If the user is evaluating an observed signal, then the same object for evaluation can be used for seeding, so long as the user specifies a cut date in plane_seed().

With the signal to be evaluated and the seed prepared, the user can run a wrapper function to generate PLANES results. For convenience, the package includes a single function (plane_score()) that wraps individual plausibility analysis functions (e.g., plane_shape(), plane_diff(), etc.) and runs all components with equal weights by default. Users can optionally customize this behavior to exclude certain components or adjust the weight that individual components receive in the overall score. As described above, certain components only apply to forecast evaluation, and as such, those are automatically excluded if an observed signal is assessed with plane_score().

Beyond the programmatic API, the rplanes package also provides a user interface (UI) to conduct PLANES analyses. The UI is delivered as a Shiny app within the package, which can be launched via the rplanes_explorer() function. Steps for preparing signal objects and seeding baseline characteristics are translated to point-and-click equivalents. The app also includes an option to use built-in example data to facilitate demonstrations and exploration of the analysis outputs. PLANES results are available in tabular format and as a series of data visualizations, including plots of specific components to visually investigate cases when plausibility flags are raised.

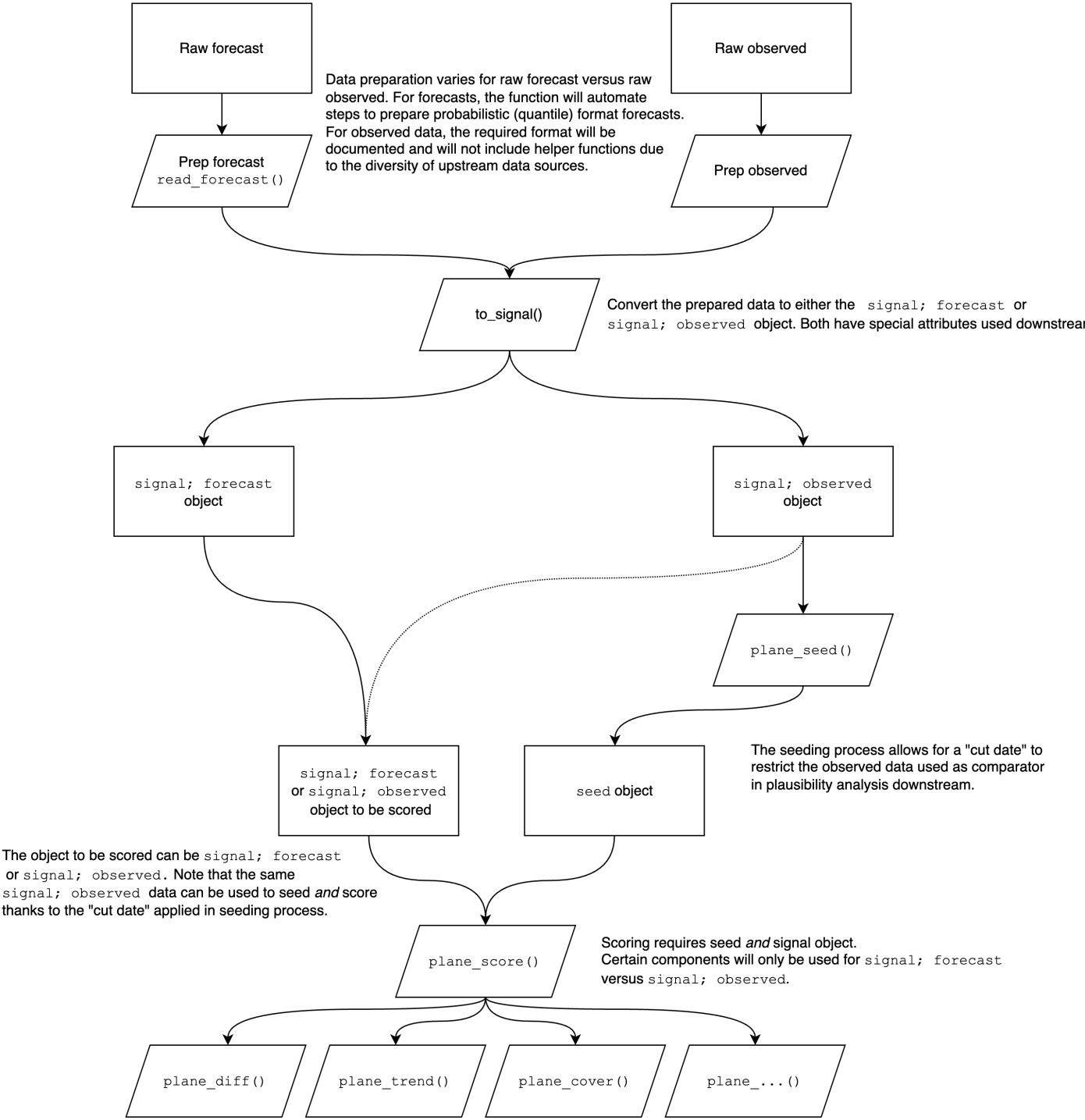

**Fig 1. Workflow for the rplanes API.** The diagram depicts the process for preparing and analyzing data with the available functions in the package. Users begin by preparing data to evaluate as well as data to seed background characteristics. These datasets begin as data frames and are coerced to 'signal' objects using rplanes helper functions. If the signal to be evaluated is observed data, then the same object used for the seed can be used in downstream analysis. If the signal is a forecast, then the user must prepare both an observed and forecast signal object for eventual scoring. The package also includes a function to build the seed using an observed signal. Given a seed and a signal to evaluate, the user can assess all components independently across all locations in the signal using a built-in wrapper function.

## Demonstration analysis

We conducted a retrospective analysis of FluSight forecasts to demonstrate the PLANES method as delivered in the rplanes package with default arguments. After retrieving all Flu-Sight submissions for the 2022-23 season, we found that 44 forecasters submitted at least one forecast between the first submission date (October 17, 2022) and final date (May 15, 2023). Of these, we evaluated 32 forecasters that were designated as either "primary" or "secondary" methods and had unambiguous, permissive licensing in the available metadata. When aggregated across all forecasters, forecast weeks, horizons, and locations, there were 120,539 forecasts analyzed. There were 53 combinations of locations, forecasters, and forecast weeks that did not include a forecast for the 4 week-ahead horizon. Additionally, there were 2,640 forecasts for Puerto Rico and Virgin Islands that were excluded prior to PLANES and WIS analysis. After aggregating across horizons, we had combined PLANES and WIS results for 29,488 forecasts.

In Fig 2, we present the number of times each component was flagged in the FluSight forecasts that were analyzed. Taper and shape were the most commonly flagged components, while difference and zero were the least common. Fig 3 shows the results of the plausibility analysis and forecast performance using the PLANES and WIS metrics respectively. The violin plots of WIS distribution are aggregated by number of PLANES components flagged in the given forecasts. The maximum number of flags raised for any given forecast was five. Roughly 40% (n = 11,738) of forecasts had no flags raised, while about another 40% (n = 11,750) had one flag raised. These forecasts had similar log-transformed median WIS values. As the number of PLANES flags increased from two through five, the median WIS appears to clearly step up in a linear pattern. After filtering for results with at least two PLANES components flagged, we found that PLANES and WIS were significantly correlated ($r = 0.267; p < 0.0001$). When assessing the impacts of the PLANES algorithm weights and components used, we found that the overall score was sensitive to these parameters. Varying the weighting schemes shifted statistical summaries of PLANES scores across all forecasters and locations. S2 Fig shows that the threshold for the 95th percentile of scores varies by forecast week and weighting scheme used. Likewise, we found that with equal weights the proportion of forecasts with at least two flags raised varied widely across different combinations of component sets. S3 Fig shows these results for all possible sets with between five and seven components. Some of the sets with fewer possible components had a higher proportion of at least two flags raised.

## Discussion

Epidemiological forecasting and reporting instruments are subject to occasionally implausible signals. Regardless of the underlying causes, having an efficient and interpretable method to review forecasts and surveillance data can help public health stakeholders address these data integrity issues in a timely manner. PLANES is a data-driven, multi-dimensional approach that was designed to fill this need. We have developed and delivered PLANES via the rplanes R package. Several groups have released methods and tools to identify data integrity issues via anomaly detection [17,18]. However, existing approaches are generally limited to investigating individual features of the time series. With PLANES we have aspired to incorporate multiple components that check for explicit plausibility expectations tailored to epidemiological signals. To our knowledge, this provides a novel method, which can readily be evaluated and incorporated into operational workflows using rplanes.

In our analysis of FluSight forecasts we found that the PLANES score was correlated with forecast performance, with forecasts scored as more implausible being more inaccurate on average. This result suggests practical utility of the PLANES approach in forecasting

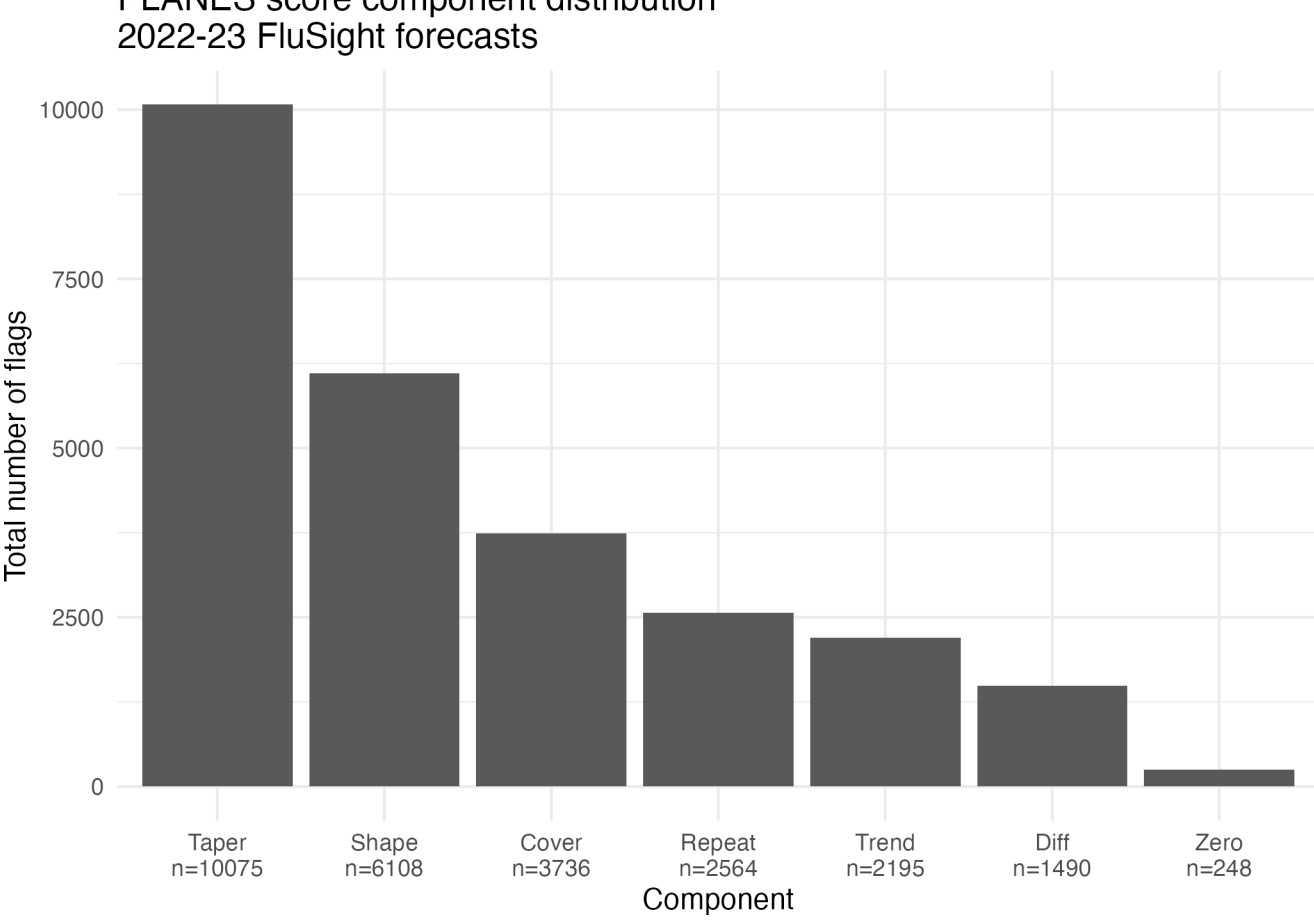

**Fig 2. Distribution of components flagged in analyzed forecasts.** The barplot shows the number of times each component was assessed as implausible in the 2022-23 FluSight forecast data. The most common components flagged were for the prediction interval tapering and novel shape, while the least common were for point-to-point difference and sudden zeros.

endeavors. For example, PLANES could be operationalized in consortia efforts to add a weighting scheme for contributed forecasts. In this use-case, forecasts submitted with higher PLANES scores may turn out to be less accurate and therefore would negatively impact the ensemble. These forecasts could be given a lower weight when creating the ensemble. Optimizing ensemble forecasting efforts is an active area of research, and PLANES provides a practical alternative to current approaches [19]. However, it is important to note that weighting schemes have been studied before and their benefit over equally weighted ensembles has not clearly been demonstrated [20].

As another practical forecasting application, the plausibility scoring could guide review and potential censoring of model output prior to submission. To encourage transparency, we recommend that users who may be censoring operational forecasts review all guidelines and policies provided by the downstream forecast consumer. Many forecasting hubs allow users to specify methods via submitted metadata files, which may be a suitable mechanism to describe any PLANES review or censoring protocol. As appropriate, specific targets or locations for which forecasts are censored should be documented along with PLANES scoring results used to guide this decision. In our analyses using PLANES with FluSight forecasts, we found that the overall plausibility score was sensitive to the weighting scheme as well as the

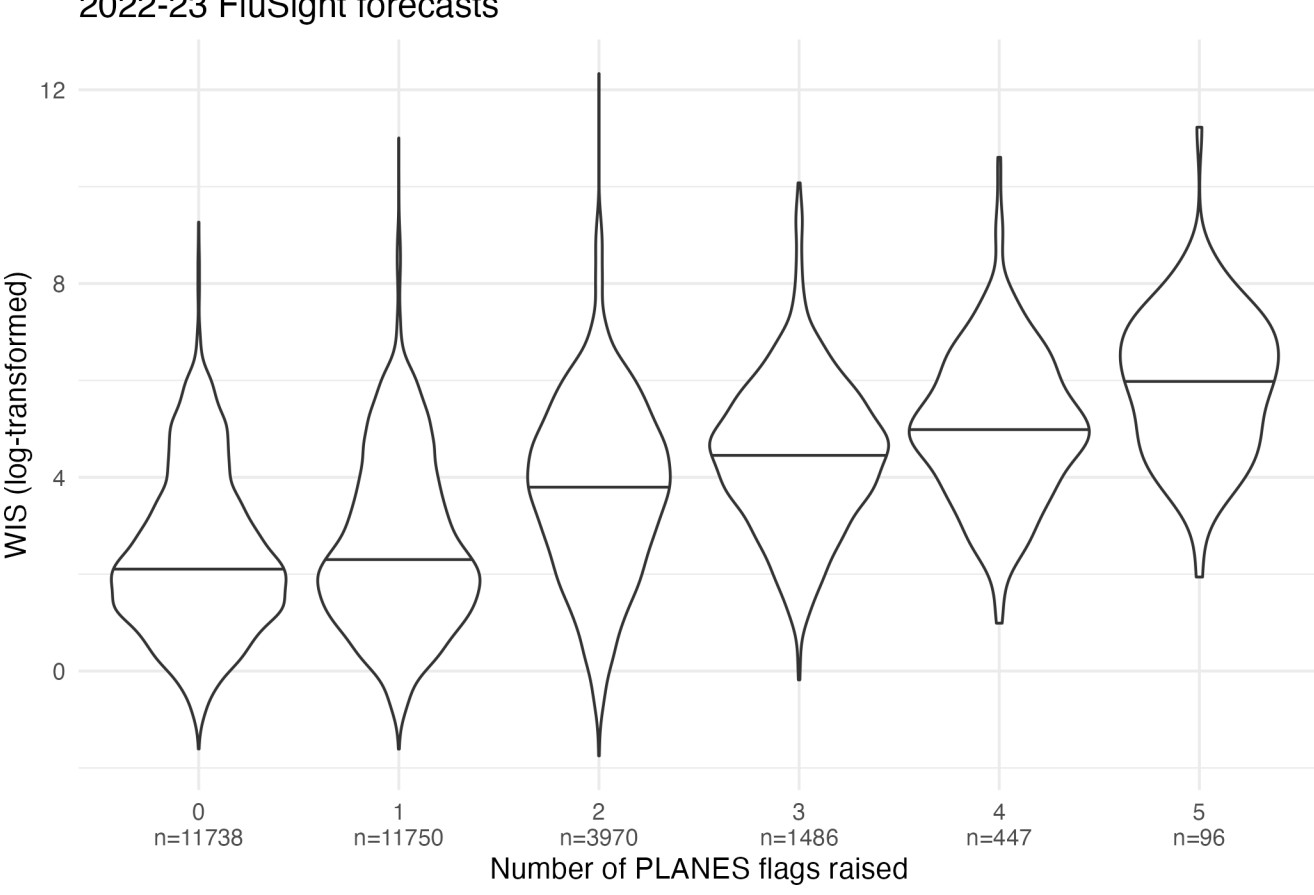

**Fig 3. Relationship between WIS and PLANES scoring.** The violin plot shows distribution of the WIS stratified by number of PLANES flags raised for the given forecast. Horizontal lines indicate median of the log-transformed WIS. The median performance of forecasts with zero or one component flagged was about equal. When forecasts had at least two PLANES flags raised, the median WIS increased (i.e., performance degraded) in a linear fashion.

set of components used. The rplanes package makes it trivial to adjust these parameters. We recommend that, to the extent feasible, users conduct retrospective analyses to understand the distribution of components flagged and overall scores over time. In doing so, we suggest testing different weighting schemes and sets of components to be included as these may influence the interpretation of overall PLANES results.

It is also worth emphasizing that PLANES may improve forecasting performance before forecasts are even created. The integrity of surveillance data used to train models can contribute to the quality of downstream forecasts. The flexibility of the PLANES methods to analyze observed signals enables public health stakeholders to review surveillance data and improve instruments that generate data used by operational modelers.

Beyond forecasting initiatives, reviewing surveillance data integrity can improve how public health data are disseminated and analyzed. While we acknowledge that we have developed our plausibility analysis approach around epidemiological practices used in the United States, PLANES is designed to be applicable in diverse surveillance contexts. We expect it can be used with data coming from surveillance systems in other regions, including low and middle-income countries, so long as the data are reported at a regular cadence. Maintaining

surveillance systems needed for successful outbreak response can be challenging in areas with limited infrastructure [21], and rplanes cannot address some of the existing constraints (e.g., no historical reporting). However, reviewing the signals that are present in these contexts can help flag other problems (e.g., surveillance signal suddenly dropping out). Furthermore, we do not anticipate that implementing rplanes would inherently present computing challenges in areas with limited infrastructure. The package is free to use and adapt, easy to install, and able to be run offline on commodity computing systems (i.e., no specialized hardware required).

We have piloted rplanes for forecast and observed signal use-cases in operational forecasting activities. For example, during the 2023-24 FluSight season we reviewed our submitted forecasts using a threshold informed by the FluSight forecast analysis presented here. Any forecasts with at least two of the PLANES components flagged were set aside for further human review. If our team deemed it necessary based on a combination of the scores and visual inspection, we removed forecasts for the given location from the weekly submission. Our forecasts were generated using models trained on flu admission data reported at the state level via the National Healthcare Safety Network (NHSN). During the 2023-24 season, the NHSN data were aggregated weekly at the state level. Using rplanes as a dependency, we built a custom API that queried the NHSN instrument and scored the data as an observed signal. Our API was automated such that it ran on a schedule and delivered an email with any flags that were raised for the weekly hospitalization data in each state. If the state-level reporting had at least one PLANES component flagged, then we manually reviewed the reported signal before fitting our models to confirm trust in the training data.

One of the key features of PLANES is that the algorithm is not fully automated. We have intentionally designed an approach that stops short of automatically intervening in data flagged as implausible (e.g., automatic censoring). Ultimately, the decision for action on a plausibility assessment will be context-specific and may necessitate human review of flagged components. The goal of the PLANES tooling is to dramatically reduce the burden of this review, and to codify metrics for more standardized interpretation and communication around signal data integrity. However, users who are interested in achieving more automated procedures will be able to build such systems around the PLANES components as they see fit. For example, if automation and human intervention becomes imbalanced (i.e., it is taking too much time to review and act on PLANES scores manually) then it is possible to use the outputs of PLANES to trigger user-defined automated censoring or correction managed outside of the rplanes functionality.

The system we have developed is intentionally modular. We have created rplanes such that the existing components can be re-weighted or ignored altogether depending on the use-case. It is important to emphasize that the set of plausibility components we have identified is based on intuition gained from our operational forecasting efforts. For example, during the 2022-23 FluSight season the forecasting community noticed that one state had the exact same hospitalization values reported for more than seven days in a row, which inspired our implementation of the repeat component. We expect that the algorithm we implemented may benefit from other plausibility heuristics. While we are not actively developing new components currently, the rplanes API is built to support new components in the future, with very minimal change to the user experience or interpretation of the PLANES scoring output.

We see the PLANES methodology and tooling as filling an immediate need in the disease modeling and epidemiological surveillance domains. However, we anticipate that there are multiple ways in which the approach may evolve and improve over time. As noted above, the modularity of PLANES invites the addition of new components. While we expect they will apply to percentages and proportions, many of the current components were developed with count data in mind. We also acknowledge that while conceiving of PLANES, we

were primarily focused on respiratory disease signals. We have used the approach to explore select vector-borne disease signals, and we expect that PLANES will require minimal (if any) adaptation for other non-respiratory signals reported at either daily, weekly, or monthly cadence. However, additional applications for non-respiratory signals may reveal new areas of improvement to the underlying algorithms.

We also acknowledge that PLANES fundamentally depends on having trustworthy historical reporting from which you can create seed characteristics. This may impact use-cases for PLANES where historical data is unavailable or sparse. Ideally, PLANES would be used when complete data (i.e., no missing observations in the time series) is available to seed background plausibility characteristics. In practice, users may encounter data that have missing observations or uncorrected historical anomalies. If data are missing from the input time series, the PLANES algorithm as currently implemented in rplanes will execute and trigger a warning message notifying the user. However, the PLANES algorithm does not adjust for sparse or unreliable data internally. Any imputation or correction to the data used to seed the plausibility analysis must be performed before running rplanes at the discretion of the user. It also worth noting that if there are gaps in the seed data, the package functions will issue a warning while proceeding with the analyses. Therefore, while it may introduce challenges in data preparation and interpretation, having sparse or incomplete data does not preclude using the PLANES approach.

Our goal in codifying the PLANES methods and building the rplanes R package was to invite users to adopt this approach as they see fit. We expect that community input will identify limitations and new applications. We openly invite feedback and contributions.

## Conclusion

We have developed a novel approach to conduct plausibility analysis of epidemiological signals. PLANES is delivered in rplanes, which is an open-source R package. In demonstration analyses, we have shown that the PLANES scoring system correlates with forecast performance. We anticipate that the methods and tooling described here will be useful to public health stakeholders, including disease modelers as well as other users and maintainers of epidemiological surveillance data systems.

## Supporting information

**S1 Fig. PLANES concept.** Illustration of the conceptual motivation for developing the PLANES approach. Several examples of possible components and their impacts are described. (TIFF)

**S2 Fig. PLANES score weighting schemes.** Demonstration of the sensitivity of PLANES scoring to varying weighting schemes. All eligible FluSight forecast submissions in the 2022-23 season were scored using PLANES under 100 randomly sampled weighting schemes. For each forecast week, the scores were aggregated across all forecasters and locations and summarized to identify the 95th percentile (q95). The tile plot is shaded by q95 at each week under the 100 different weighting schemes. At certain points in the season scores are consistently higher or lower across weighting schemes. For example, PLANES scores for late season forecasts have a lower q95. However, the weighting schemes visibly shift the q95 threshold for many forecast weeks. (TIFF)

**S3 Fig. PLANES score component combinations.** Impact of PLANES component sets on overall scoring. All eligible FluSight forecast submissions in the 2022-23 season were scored

using all combinations of PLANES components. Across all forecast weeks, forecasters, and locations, we counted the number of PLANES scores with at least two flags raised using each component set. In this figure, the results are restricted to component sets with between five and seven components. Within sets that had the same overall number of possible components, there was a wide range in the proportion of scores that had at least two flags raised. For example, the proportion ranged between 0.06 and 0.19 among sets with five components. In some cases, sets with fewer components had a higher proportion.
(TIFF)

**S1 Table. Technical limitations, mitigations, and future enhancements.** Selection of technical limitations along with their current mitigations and relevant potential enhancements that could be made in the future.
(XLSX)

**S1 File. Additional details on PLANES components.** Equations and examples for each of the seven PLANES components.
(PDF)

## Acknowledgments

The work described in this manuscript would not have been possible without open and collaborative efforts from multiple entities. We acknowledge the following groups: the Armed Forces Health Surveillance Division - Integrated Biosurveillance Branch (AFHSD-IB) for their willingness to provide continued feedback on the plausibility approach while piloting the rplanes R package; the CDC for coordinating FluSight and providing guidance, interpretation, and dissemination of forecast data; the Council of State and Territorial Epidemiologists (CSTE) for establishing collaborative networks through which forecasting groups can interact with one another and public health stakeholders; all participating teams in the FluSight network for their sustained contributions, innovative techniques, and commitment to openness through operational forecasting activities.

## Author contributions

**Conceptualization:** V.P. Nagraj, Stephen D. Turner.

**Data curation:** V.P. Nagraj.

**Formal analysis:** V.P. Nagraj, Amy E. Benefield.

**Funding acquisition:** V.P. Nagraj.

**Investigation:** V.P. Nagraj, Amy E. Benefield.

**Methodology:** V.P. Nagraj, Amy E. Benefield.

**Project administration:** V.P. Nagraj.

**Resources:** V.P. Nagraj.

**Software:** V.P. Nagraj, Amy E. Benefield, Desiree Williams.

**Supervision:** V.P. Nagraj.

**Validation:** V.P. Nagraj.

**Visualization:** V.P. Nagraj, Desiree Williams.

**Writing – original draft:** V.P. Nagraj.

**Writing – review & editing:** V.P. Nagraj, Amy E. Benefield, Desiree Williams.

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
