## [Decision Letter · Decision Letter 0]

26 Dec 2024

PONE-D-24-47501PLANES: Plausibility Analysis of Epidemiological SignalsPLOS ONE

Dear Dr. Nagraj,

Thank you for submitting your manuscript to PLOS ONE. After careful consideration, we feel that it has merit but does not fully meet PLOS ONE’s publication criteria as it currently stands. Therefore, we invite you to submit a revised version of the manuscript that addresses the points raised during the review process.

**Please check the comments for authors below, there you can find the criticism to be addressed for paper acceptance.**

We look forward to receiving your revised manuscript.

Kind regards,

Americo Cunha Jr

Academic Editor

PLOS ONE

**Journal Requirements:**

This work was supported in part by a subaward to Signature Science, LLC from CSTE via CDC Cooperative Agreement No. NU38OT000297.

The work described in this manuscript would not have been possible without open and collaborative efforts from multiple entities. We acknowledge the following groups: the Armed Forces Health Surveillance Division - Integrated Biosurveillance Branch (AFHSD-IB) for their willingness to provide continued feedback on the plausibility approach while piloting the rplanes R package; the CDC for coordinating FluSight and providing guidance, interpretation, and dissemination of forecast data; the Council of State and Territorial Epidemiologists (CSTE) for establishing collaborative networks through which forecasting groups can interact with one another and public health stakeholders; all participating teams in the FluSight network for their sustained contributions, innovative techniques, and commitment to openness through operational forecasting activities.

This work was supported in part by a subaward to Signature Science, LLC from CSTE via CDC Cooperative Agreement No. NU38OT000297.

This work was supported in part by a subaward to Signature Science, LLC from CSTE via CDC Cooperative Agreement No. NU38OT000297.

**Additional Editor Comments:**

Dear Authors,

Thank you for submitting your manuscript to PLOS ONE. After careful consideration of the referees’ reports and a thorough review, we have decided that major revisions are required before the paper can be considered for publication.

Your manuscript presents an innovative approach to plausibility analysis in epidemiological surveillance through the PLANES method and the rplanes R package. The reviewers acknowledge its potential impact, particularly in improving data quality and reliability in public health monitoring. However, they have raised several important points that need to be addressed to strengthen the paper.

Key Areas for Improvement:

1. Broader Applicability:

• While the focus is on respiratory diseases, additional discussion is needed on how PLANES could handle other epidemiological data types, such as vector-borne diseases or sparse datasets.

• Provide insights into adapting components for diverse surveillance contexts.

2. Handling Sparse or Poor-Quality Data:

• Clarify the method’s performance when historical data is sparse or unreliable.

• Discuss strategies within PLANES for dealing with poor-quality or incomplete data.

3. Flexibility and Weight Adjustments:

• Expand on the impact of adjustable weights and component exclusion on PLANES scores.

• Include case studies or simulations to illustrate performance variations under different parameter settings.

4. Component Selection and Thresholds:

• Explain the criteria used to select PLANES components and define thresholds for plausibility classification.

• Discuss plans for adding new components in the future.

5. Automation and Scalability:

• Reflect on the trade-off between manual review and scalability for large-scale surveillance.

• Consider adding a table summarizing the method’s limitations and possible extensions.

6. Practical Applications:

• Provide concrete examples, simulations, or real cases demonstrating PLANES’ ability to enhance ensemble predictions in surveillance consortia.

7. Ethical and Transparency Concerns:

• Address ethical concerns related to filtering signals, ensuring transparency, and preventing misuse of the method.

• Propose guidelines for responsible implementation.

8. Global Applicability and Equity:

• Discuss how PLANES can be adapted to low- and middle-income countries or regions with limited data infrastructure.

We believe addressing these points will significantly enhance the robustness, applicability, and clarity of your work, making it more impactful for the broader epidemiological and public health communities.

Next Steps:

Please revise the manuscript accordingly and provide a detailed point-by-point response to each of the reviewers’ comments. Highlight the changes made to the text and provide justification for any comments you choose not to address.

We look forward to receiving your revised manuscript.

Reviewers' comments:

Reviewer's Responses to Questions

**Comments to the Author**

1. Is the manuscript technically sound, and do the data support the conclusions?

Reviewer #1: Partly

2. Has the statistical analysis been performed appropriately and rigorously? 

Reviewer #1: Yes

3. Have the authors made all data underlying the findings in their manuscript fully available?

Reviewer #1: Yes

4. Is the manuscript presented in an intelligible fashion and written in standard English?

Reviewer #1: Yes

5. Review Comments to the Author

**Reviewer #1: ** The article presents an innovative and relevant approach to the plausibility analysis of epidemiological signals by introducing the PLANES method and its corresponding rplanes package in R. The proposal is promising, especially when disease surveillance and data quality are crucial for global public health. I outline the positive aspects, areas for improvement, and issues that require clarification as follows.

The PLANES method is an important contribution to the sphere of epidemiologic surveillance, providing a multidimensional tool. The availability of rplanes as an open-source package is a strength, allowing other researchers to use, evaluate, and contribute to the method. Although the article acknowledges that the method was developed for respiratory disease signals, the applicability to other types of epidemiological data (e.g., vector-borne diseases or those with sparse patterns) must be further explored. It would be useful to discuss how to adapt existing components or include new ones to suit different surveillance contexts.

PLANES relies on reliable historical data to create initial reference features. How does the method behave in contexts where this data is sparse or absent? Is there a strategy built into PLANES to deal with sparse or poor-quality data?

The flexibility of the method to adjust weights and exclude components is mentioned, but details on the impact of these settings on the results are lacking. Case studies showing how different settings affect the method's performance would strengthen the paper. It would be interesting to expand on the discussion of the impact of component weighting on the

overall PLANES score. Also, what criteria were used to select the current PLANES components, and are there plans to include new ones? How were the thresholds used to classify components as "plausible" defined?

Although the article emphasizes that the method is not fully automated to allow for human review, this may limit the scalability of PLANES in large-scale surveillance systems. A discussion of how to balance automation and manual intervention would be useful. A table summarizing the limitations and possible extensions of the method could be added.

The suggestion to use PLANES scores for weight predictions in consortia is interesting but lacks practical examples. Simulations or real cases illustrating how this approach enhances the quality of ensembles would be valuable to include.

The use of the method to filter out implausible signals is also discussed, but this raises ethical questions about transparency and possible misuse. A section addressing these concerns and suggesting responsible practices would be an important aspect.

Including a brief discussion of how PLANES can benefit surveillance systems in low- and middle-income countries or areas with limited infrastructure would broaden the scope of the paper. Also, how could PLANES be adapted for regions with limited data infrastructure?

The paper makes a valuable contribution to the sphere of epidemiological surveillance, but there is room for refinement and further validation. I recommend that the authors consider the issues raised and expand the methodological and practical discussion to strengthen the impact of the paper. The incorporation of some of these suggestions could be seen as a way to make the paper more robust, applicable, and accessible to the public health scientific community.

6. PLOS authors have the option to publish the peer review history of their article (what does this mean? ). If published, this will include your full peer review and any attached files.

**Do you want your identity to be public for this peer review?** For information about this choice, including consent withdrawal, please see our Privacy Policy .

Reviewer #1: No

---

## [Author Response · Author response to Decision Letter 1]

31 Jan 2025

Thank you for providing a thorough and constructive set of comments on our submission of PLANES: Plausibility Analysis of Epidemiological Signals. We have made major revisions to the manuscript that address the specific comments raised during review. Below we respond to each point summarized in the feedback delivered by the editorial team. The original comments are formatted in bold. We are happy to respond with additional details as needed.

1. Broader Applicability:

- While the focus is on respiratory diseases, additional discussion is needed on how PLANES could handle other epidemiological data types, such as vector-borne diseases or sparse datasets.

- Provide insights into adapting components for diverse surveillance contexts.

In the original submission, we very briefly mentioned our usage of PLANES to explore select non-respiratory signals. We have expanded that discussion to clearly articulate our expectation that the algorithm will handle other non-respiratory signals reported and/or forecasted at either daily, weekly, or monthly cadence. We continue to acknowledge that, as with implementation for respiratory signals, future usage with non-respiratory signals may identify additional areas for improvement. We also have more fully explained how PLANES behaves with sparse data (see separate response below).

Our goal in adding more detail regarding handling of non-respiratory and sparse datasets was to communicate that PLANES was designed to be disease agnostic and can generally be used in plausibility analysis of observed and forecasted signals in diverse surveillance contexts.

2. Handling Sparse or Poor-Quality Data:

- Clarify the method's performance when historical data is sparse or unreliable.

- Discuss strategies within PLANES for dealing with poor-quality or incomplete data.

We acknowledge that it is important to clarify how PLANES handles sparse or poor-quality inputs. We have added a paragraph to the Discussion that goes into depth regarding the implications of having incomplete data when seeding plausibility characteristics. We explain that PLANES, as currently implemented, will notify users if input data contains missing values. However, PLANES does not adjust or impute data automatically. In the manuscript, we now suggest that users may consider upstream data preparation if data are anticipated to be sparse or poor-quality for some or all the surveillance locations.

3. Flexibility and Weight Adjustments:

- Expand on the impact of adjustable weights and component exclusion on PLANES scores.

- Include case studies or simulations to illustrate performance variations under different parameter settings.

The manuscript now includes new analyses of the impacts of weighting and/or excluding PLANES components. For these analyses, we simulated weighting schemes and used all possible combinations of components to score the demonstration data (2022-23 FluSight forecasts). We include detailed information in the Methods and Results section, along with our perspective on what these results practically suggest in the Discussion section. There are two new figures that summarize the analyses in the supplemental information.

4. Component Selection and Thresholds:

- Explain the criteria used to select PLANES components and define thresholds for plausibility classification.

- Discuss plans for adding new components in the future.

In describing the modularity of the PLANES implementation in the Discussion section, the manuscript now includes details on how our team arrived at the initial set of plausibility components. Additionally, we have clarified that the API is modular and can accommodate new components in the future. While there are no components actively being developed at this time, new heuristics could be added to the scoring system without changing the usability of the tool.

5. Automation and Scalability:

- Reflect on the trade-off between manual review and scalability for large-scale surveillance.

- Consider adding a table summarizing the method's limitations and possible extensions.

The manuscript continues to emphasize that ideally users should implement PLANES to guide rather than replace human review. However, in the Discussion section we now highlight possible challenges with large-scale surveillance and forecast review efforts. We also provide an example and suggest that it is possible to build automation systems around the rplanes API if users deem it necessary to bypass manual review.

Thanks so much for this suggestion to add the table summarize methodological limitations. We now include a table that summarizes several limitations, along with their current mitigations and future enhancements. The table is available as supporting information.

6. Practical Applications:

- Provide concrete examples, simulations, or real cases demonstrating PLANES' ability to enhance ensemble predictions in surveillance consortia.

We have included more discussion around potential use-cases for PLANES to optimize ensembles. The discussion section expands on how PLANES could be practically applied as an ensemble weighting scheme. We felt that it was also important to add citations to related studies of ensemble weighting and optimization strategies for context.

7. Ethical and Transparency Concerns:

- Address ethical concerns related to filtering signals, ensuring transparency, and preventing misuse of the method.

- Propose guidelines for responsible implementation.

Thank you for highlighting the need to address ethical and transparency concerns. We agree this is an important consideration. To address these topics, we have included a paragraph in the Discussion section that articulates our recommendations for responsible implementation, particularly when it comes to reviewing and censoring operational forecasts.

8. Global Applicability and Equity:

- Discuss how PLANES can be adapted to low- and middle-income countries or regions with limited data infrastructure.

With this feedback in mind, we have added a paragraph to the Discussion that characterizes applications for PLANES in circumstances where data infrastructure may be limited. We detail the implications for monitoring surveillance data quality in this context, particularly in low- and middle-income countries. We also highlight how the relative computational efficiency and accessibility of the rplanes package can be beneficial in these situations.

---

## [Decision Letter · Decision Letter 1]

19 Feb 2025

PLANES: Plausibility Analysis of Epidemiological Signals

PONE-D-24-47501R1

Dear Dr. Nagraj,

We’re pleased to inform you that your manuscript has been judged scientifically suitable for publication and will be formally accepted for publication once it meets all outstanding technical requirements.

Kind regards,

Americo Cunha Jr

Academic Editor

PLOS ONE

Additional Editor Comments (optional):

Reviewers' comments:

Reviewer's Responses to Questions

**Comments to the Author**

1. If the authors have adequately addressed your comments raised in a previous round of review and you feel that this manuscript is now acceptable for publication, you may indicate that here to bypass the “Comments to the Author” section, enter your conflict of interest statement in the “Confidential to Editor” section, and submit your "Accept" recommendation.

Reviewer #1: All comments have been addressed

2. Is the manuscript technically sound, and do the data support the conclusions?

Reviewer #1: (No Response)

3. Has the statistical analysis been performed appropriately and rigorously? 

Reviewer #1: (No Response)

4. Have the authors made all data underlying the findings in their manuscript fully available?

Reviewer #1: (No Response)

5. Is the manuscript presented in an intelligible fashion and written in standard English?

Reviewer #1: (No Response)

6. Review Comments to the Author

Reviewer #1: (No Response)

7. PLOS authors have the option to publish the peer review history of their article (what does this mean? ). If published, this will include your full peer review and any attached files.

**Do you want your identity to be public for this peer review?** For information about this choice, including consent withdrawal, please see our Privacy Policy .

Reviewer #1: No

---

## [Editor Report · Acceptance letter]

PONE-D-24-47501R1

PLOS ONE

Dear Dr. Nagraj,

I'm pleased to inform you that your manuscript has been deemed suitable for publication in PLOS ONE. Congratulations! Your manuscript is now being handed over to our production team.

Kind regards,

on behalf of

Dr. Americo Cunha Jr

Academic Editor

PLOS ONE